

# Advanced multiple document summarization *via* iterative recursive transformer networks and multimodal transformer

Sunilkumar Ketineni and Sheela Jayachandran

SCOPE, VIT-AP University, Amaravathi, Andhra Pradesh, India

## ABSTRACT

The proliferation of digital information necessitates advanced techniques for multiple document summarization, capable of distilling vast textual data efficiently. Traditional approaches often struggle with coherence, integration of multimodal data, and suboptimal learning strategies. To address these challenges, this work introduces novel neural architectures and methodologies. At its core is recursive transformer networks (ReTran), merging recursive neural networks with transformer architectures for superior comprehension of textual dependencies, projecting a 5–10% improvement in ROUGE scores. Cross-modal summarization employs a multimodal transformer with cross-modal attention, amalgamating text, images, and metadata for more holistic summaries, expecting an 8 to 12% enhancement in quality metrics. Actor-critic reinforcement learning refines training by optimizing summary quality, surpassing Q-learning-based strategies by 5–8%. Meta-learning for zero-shot summarization addresses summarizing unseen domains, projecting a 6–10% uptick in performance. Knowledge-enhanced transformer integrates external knowledge for improved semantic coherence, potentially boosting ROUGE scores by 7 to 12%. These advancements not only improve numerical performance but also produce more informative and coherent summaries across diverse domains and modalities. This work represents a significant stride in multiple document summarization, setting a new benchmark for future research and applications.

Corresponding author
Sheela Jayachandran,
sheela.j@vitap.ac.in

## INTRODUCTION

In the digital era, the exponential growth of textual data across diverse domains presents a formidable challenge for information retrieval and comprehension. The need for efficient summarization techniques that can condense, extract, and articulate the core messages from voluminous documents is more pressing than ever. Multiple document summarization, a complex task aimed at generating concise and coherent summaries from a collection of documents, stands at the forefront of addressing this challenge. Traditional summarization methods, while foundational, have increasingly shown limitations in grappling with the

intricacies of processing and integrating vast and varied textual information. This paper introduces groundbreaking advancements in multiple document summarization, heralding a paradigm shift towards more sophisticated, accurate, and versatile summarization capabilities.

Historically, summarization techniques have primarily focused on single-document summarization, with multiple document summarization receiving comparatively less attention. The complexity of understanding, cross-referencing, and condensing information from multiple sources poses unique challenges, including the maintenance of coherence, avoidance of redundancy, and the integration of diverse viewpoints. Additionally, the advent of multimodal content, combining text, images, and metadata, further complicates the summarization task, demanding innovative approaches that can seamlessly synthesize information across different modalities.

Addressing these challenges, this work introduces an array of advanced neural architectures and methodologies designed to significantly enhance the efficiency and effectiveness of multiple document summarization. Central to our contribution is the development of recursive transformer networks (ReTran). This ingeniously combine the hierarchical processing capabilities of recursive neural networks with the dynamic representation learning of transformer architectures. This integration allows for an unprecedented understanding of local and global dependencies within texts, setting the stage for summaries that are not only accurate but also contextually rich and coherent (*Yadav et al., 2024*; *Yadav et al., 2023*).

Expanding the horizon to multimodal data, we present the multimodal transformer with cross-modal attention, a pioneering method that adeptly merges textual and non-textual information to produce summaries that reflect a more comprehensive grasp of the content. This approach underscores the significance of cross-modal information processing in enhancing the depth and informativeness of summaries.

In pursuit of optimizing summarization quality, our exploration into deep reinforcement learning, specifically through actor-critic reinforcement learning for summary generation, offers a robust framework for training summarization models. By directly optimizing the quality of summaries and ensuring a stable learning trajectory, this method marks a significant improvement over traditional reinforcement learning strategies.

Moreover, the paper delves into zero-shot and few-shot learning techniques, particularly meta-learning for zero-shot summarization, to tackle the challenge of summarizing documents from unseen domains or with minimal training data samples. This approach demonstrates our commitment to versatility and adaptability in summarization tasks.

Lastly, our work introduces the knowledge-enhanced transformer for summarization, a method that leverages external knowledge bases to imbue summaries with semantic coherence and domain-specific insights. This method not only advances the state-of-the-art in summarization but also paves the way for integrating semantic reasoning and commonsense knowledge into the summarization process.

From efficiency and coherence to adaptability, there is much to the limitations of current state-of-the-art methods for multi-document summarization. Whereas the traditional models in MDSS—especially those with a basic architecture of transformer—have proven

unmatched by any other model for capturing long-range text dependencies through self-attentive mechanisms, they often miss out on modeling hierarchies in documents that are complex. Such a weakness is more pronounced in multi-document summarization tasks, where the relationships between diverse sources of content need to be comprehended both at granular and holistic levels. In this respect, while models like BART and PEGASUS improved generation capabilities, they produce summaries that are a little lacking in depth concerning coherence and semantic understanding across varied document structures. Also, many of these models target vision or language alone for single-modality inputs, which reduces their application scope in reality when multimodal data-continuously textual with image or metadata-is becoming more and more relevant.

Another key limitation of the current methods is that most of them require massive amounts of domain-specific labeled data; therefore, applying these methods in zero-shot or few-shot learning is rather tricky. Unfortunately, the models based on supervised learning are somewhat good at generalization and train on large datasets, hence failing to execute well in new domains with limited labeled data and thus leading to suboptimal performance. Apart from generalization, these models usually do not incorporate any external knowledge such as domain-specific knowledge graphs or ontologies, which is one of the most important requirements for generating a summary that is not only accurate but semantically rich and informative. In addition, most summarization models employ Q-learning-based reinforcement learning techniques for the sake of optimizing summary quality. However, most of these techniques face a critical challenge of striking between summary conciseness and its relevance and coherence, thus coming up with a summary that either lacks critical information or includes redundant and less relevant information sets.

Motivated by the above limitations, the paper introduces various new techniques boosting the quality of multiple document summarizations. The approach was selecting ReTran in the view of making up the deficiency of a hierarchical process by using recursive neural networks in conjunction with the transformer architecture. This would make it possible to create a model that will substantially represent both local and global dependencies within a document and result in much more detailed knowledge about its structure. On the other hand, by explicitly modeling hierarchical relationships, ReTran overcomes the drawbacks of standard transformers that mainly focus on long-range dependencies without accounting for the inherent structure of language.

Introduction of the multimodal transformer with cross-modal attention: Thus, it addresses the limitation regarding single-modality summarization models. It combines text and images with metadata to generate comprehensive summaries representative of diverse modalities in real-world samples. Cross-modal attention allows dynamic weighting between the different types of inputting, so that the most important information of each modality is integrated into the final summary. This technique constitutes a huge improvement in contexts where visual and contextual data plays a critical role when trying to understand the content, addressing the gap left by traditional models that operate based on text inputs for the process.

This might also be considered a major step beyond Q-learning-based approaches. The model represents an attempt to automatically adjust its strategy of summarization in line with the predefined metrics of quality since the summary quality is continuously optimized in the feedback loop between the actor and critic. Hence, this approach essentially tends to enhance the model's coherence and conciseness of summaries, relevant-inherent weaknesses of earlier techniques that tried to balance this reinforcement learning objective.

Other key innovation is meta-learning for zero-shot summarization, which tackles the problem of generalizing poorly to unseen domains. The model will learn from the meta-training and meta-testing stages to adapt quickly to new tasks, even with few data points, and large improvement is achieved in both zero-shot and few-shot learning scenarios. This technique, without any need for great domain-specific labeled data, presents an effective solution to summarize content from new and emerging fields. Finally, the knowledge-enhanced transformer facilitates the process of summarization by baking outside knowledge right into the model architecture. Using structured knowledge, such as knowledge graphs, the model summarizes information that is semantically coherent and contextually rich. This approach supplements certain deficiencies of the basic transformer model, since these often are not deep enough to appreciate nuances in domain-specific content and hence summarize it for different scenarios.

Through these contributions, this paper not only addresses the inherent limitations of existing summarization methods but also pioneers a comprehensive framework for future research and applications in multiple document summarization. The advancements presented herein are poised to redefine the boundaries of what is achievable in automatic summarization, offering pathways to more intelligent, adaptable, and insightful summarization technologies that can keep pace with the ever-growing deluge of digital information sets.

## Discussions
### Generalizability of the model process

The generalisability of the proposed summarization model has to be tested on more diverse, and probably unseen, datasets that represent a wide variety of real-world scenarios. The existing framework already involves a number of popular datasets, such as CNN/Daily Mail, XSum, and RDF2Text; extension of evaluation to broader, more diverse datasets is very indispensable for gaining a more profound insight into what the true versatility and adaptability of the model across different domains can be, especially for unstructured or specialized ones.

Performance on unseen datasets is informative for generalizing beyond the model's capability to data types it has seen. In particular, it will enable zero-shot or few-shot learning parts inside the model to learn how to quickly adapt to new domains with little or no additional training data. While the tailored domain-specific datasets in this work provided an initial test of these abilities, they are rather narrow. Testing the model on more niche datasets drawn from domains such as financial reports, legal contracts, clinical research, and historical archives can give insights regarding how well the model can handle highly specialized vocabulary, domain-specific content, and structures that

are possibly unfamiliar to the model. Another issue not hitherto explored considers the generalizability of the model to multilingual and cross-lingual datasets. While the performance on the English datasets has been great, the real-world applications require multiple languages, particularly in the globalized industries like science, media, and business. This can be ensured by incorporating more varied multilingual data-sets, either on large-scale multilingual summarization corpora such as WikiLingua or on datasets from cross-lingual summarization tasks. It could test the model's ability to generate precise summaries in various languages or even translate summaries across languages. That would give insights into its robustness in handling diversity in syntax, cultural nuances, or otherwise.

It would also be interesting to go deeper into multimodal diversity. Although the Multi30k dataset provides a starting point for assessing multimodal competencies that join text and images together, this dataset is comparatively simple compared to naturally occurring multimodal data, whose relationships among text, images, videos, and structured metadata are often much more complex. Increasing evaluation on more challenging multimodal datasets, such as TVSum, a video summarization dataset, or How2, a multimodal dataset containing video with either text or speech transcripts, would test the model better as to how well it handles the information coming from different modalities. This could assess the multimodal transformer's ability to balance textual and visual inputs so that information deemed relevant from both sources will be synthesized appropriately in the final summary.

Testing the model with noisy and unstructured data sets may further reveal its generalizability. Real data is normally unstructured, with partial sentences, poor grammar, and nonsensical typing formats. Data sets designed to embed this level of noise, such as Amazon reviews—which are for the most part informal, filled with user-generated content—would more appropriately gauge the robustness in handling messy or unstructured data samples.

## Limitations

Although promising gains are shown to be attained with the proposed summarization model, there is a couple of limitations that can be considered and some possible biases, especially in scaling up the model to applications. It may not work quite as well when it came to very large datasets or documents with very complex structures. While good for these types of datasets, CNN/Daily Mail and Multi30k definitely contain well-structured texts; it has never been properly investigated with respect to the skills for summarizing really long documents, such as books, legal depositions, and technical manuals. Such a document is often made up of heavy cross-references, multilayered sections, and therefore dense technical details that demand nothing but summarization; they also require semantically deeper understanding. While for these, ReTran might fail to keep coherence for larger spans of a document, as their recursive process is computationally heavier with less effectiveness as document lengths increase in the process.

Further, its very reliance on multimodal and external knowledge input could lead to biases regarding data type prioritization or combination. For instance, the multimodal

transformer cross-modal attention dynamically adjusts weights to text, images, and metadata. On the other hand, if there is inequality in quality or relevance with respect to multimodal inputs-say, missing or poorly created images or incomplete metadata-the model will be partial or incomplete as well with respect to its summaries. Possible biases can be seen as overemphasizing these types of information, for example, visual data where the scenarios have preferably chosen to be more important through text and *vice versa*. In real life, this might mean such biases lead to less-than-optimal quality and accuracy of generated summaries when sources are either non-standardized or simply unreliable; hence, the results could fail to represent nuances in input content to their fullest extent. That is related to yet another concern: the model's performance in a real-life, dynamic environment. While the model has been vigorously tested on static datasets, situations one would face in the real world include live news feeds, streams from social media, and reports that are often updated. This would dynamically change the data and require a system of continuous adaptation and updating to reflect the most recent information. This might inject a recency bias within the generated summaries, whereby the most recent information leads at the expense of crucial historical context and background. Also, high computational complexity for the model-*i.e.*, when it comes to real-time applications-serious scalability will impede a combination of multimodal data with recursive processing due to huge computational resources. This may limit its deployment to resource-constrained environments, such as in mobile devices or low-latency applications, where efficiency and speed are paramount. In that respect, these limitations would then have to be addressed to make the model robust and applicable to a greater range of real-world uses.

## Complexity of the model process

Although the advanced complexity of the proposed model does indeed constitute one of the considerations that are necessary for its practical implementation and further research, the modular design of this model, where each component like ReTran, multimodal transformer with cross-modal attention, and actor-critic reinforcement learning plays a different role, allows flexibility in its implementation. Instead of reproducing the whole architecture, researchers or practitioners can extend parts to adapt to their needs. For example, ReTran could be an important component in extending hierarchical text processing, without any multimodal extensions required, whereas summary quality can be optimized using actor-critic reinforcement learning separately. Also, it leverages widely used frameworks like PyTorch and offers ease of implementation by providing pre-trained models. Therefore, while the overall model may seem daunting, its modularity and reliance on established techniques assure accessibility for practical use and future research adaptations.

## Discussion on user feedbacks

User feedback plays an important step in the iterative refinement process of the model, particularly through the use of actor-critic reinforcement learning (ACRL). It also provides the canvas for embedding feedback into the process for dynamic optimization of the quality of the generated summaries. The process begins with the actor, responsible for generating the initial summary based on input documents. The next stage is the critic, which evaluates

the summary against a set of predetermined criteria that include coherence, conciseness, and relevance-aside from the feedback from the user, if available. The critic uses these factors to produce a reward score that is indicative of the summary quality. That is then also an added, direct signal to this process if it incorporates user feedback. There could be various forms of this: either a rating on the quality of the summary or even specific suggestions where and how the thing should be improved-mostly with more emphasis on certain details or closer to the original text. By using this feedback, the critic updates the reward function, which tunes the model to be more aligned with the user's preference. In those cases where a user would provide feedback that the summary does not capture the important details in it, then the critic will penalize the actor's output, and in return, the actor modifies its strategy of summarization in the next iteration to include detailed full content. This feedback loop allows the model, through iterations, to be constantly refining its summaries for better alignment with what should be the output.

Moreover, the evaluation of the critic varies over cumulative feedback, which in turn means that the model will continuously improve with every new iteration of feedback to provide and identify user preferences. This iterative refinement makes the summaries non-fixed; instead, they dynamically change based on the dynamic needs of users and feedback. This therefore makes the results more relevant and accurate. This reinforcement learning approach is feedback-driven and integrates into this model to reach an optimum for the automation *versus* customization tradeoff. In other words, it means that the summaries generated will be of high quality but will also exactly meet the expectations of the users.

## Motivation & contribution

The motivation behind this research stems from an acute awareness of the burgeoning challenges posed by the relentless expansion of digital information sets. In today's information-saturated world, the ability to efficiently distill, comprehend, and convey the essence of voluminous textual and multimodal data is not just a convenience but a necessity. The limitations of current summarization techniques—ranging from their struggle with maintaining coherence in summaries derived from multiple documents to their inability to effectively integrate multimodal data sources—underscore the urgent need for innovative solutions. This pressing demand motivates the development of advanced neural architectures and methodologies capable of transcending the boundaries of traditional summarization approaches, thereby facilitating a leap towards more intelligent, adaptive, and comprehensive summarization technologies.

The contributions of this work are manifold and significant, directly addressing the aforementioned challenges through a series of pioneering advancements in the field of multiple document summarization:

- **Introduction of recursive transformer networks (ReTran)**: This novel architecture merges the hierarchical processing strength of recursive neural networks with the dynamic representation learning capabilities of transformer models. By doing so, ReTran adeptly captures both local and global dependencies within and across documents,

heralding a new era of summarization models capable of generating more coherent and contextually enriched summaries.

- **Development of a multimodal transformer with cross-modal attention**: Recognizing the importance of multimodal data in conveying comprehensive information, this method innovatively integrates textual and non-textual inputs (such as images and metadata) to produce summaries that reflect a deeper, more holistic understanding of the content. This approach signifies a major stride towards effectively summarizing and leveraging the wealth of information contained within multimodal data sources.

- **Advancement in deep reinforcement learning through actor-critic methods**: By employing an actor-critic framework for summary generation, this research offers a more stable and effective approach to training summarization models. This method stands out for its direct optimization of summary quality, representing a sophisticated alternative to traditional reinforcement learning strategies that often struggle with stability and performance consistency.

- **Exploration of zero-shot and few-shot learning techniques**: The application of meta-learning for zero-shot summarization addresses the critical challenge of generating summaries for documents from unseen domains or with limited training data samples. This contribution is pivotal for enhancing the adaptability and generalizability of summarization models, enabling them to perform effectively across a wide range of contexts and domains with minimal prior exposure.

- **Integration of semantic understanding through knowledge-enhanced transformers**: By incorporating external knowledge sources into the summarization process, this method significantly enhances the semantic coherence and domain-specific relevance of the generated summaries. This advancement not only improves the quality of summaries but also facilitates a deeper integration of semantic reasoning and commonsense knowledge into automatic summarization.

Together, these contributions represent a comprehensive effort to address the critical limitations of existing summarization methods and to push the boundaries of what is currently achievable in the domain of multiple document summarization. This work not only sets a new benchmark for future research and applications but also offers tangible pathways towards realizing more effective, efficient, and insightful summarization technologies capable of navigating the complexities of the digital ages.

## IN DEPTH REVIEW OF EXISTING MODELS

Recent advancements in text summarization have been driven by the need for efficient information extraction across various domains. Multi-document summarization has particularly gained attention for distilling crucial insights from document collections. *Li & Xu (2023)* introduced HierMDS, a hierarchical model that improves summarization by capturing global-local document dependencies. *Mulla & Shaikh (2024)* proposed a BiLSTM classifier, demonstrating enhanced summarization performance. *Singh, Mittal & Chouhan (2024)* employed deep learning techniques, showcasing promising results with long short-term memory (LSTM) and an improved optimizer.

Clustering-based approaches have also shown promise in enhancing multi-document summarization. *Saini et al. (2023)* utilized a multi-view multi-objective clustering framework for scientific document summarization, effectively leveraging citation context. *Al-Taani & Al-Sayadi (2024)* explored extractive text summarization of Arabic multi-documents using fuzzy clustering techniques, highlighting the effectiveness of fuzzy clustering in capturing document semantics. In the realm of single-document summarization, various innovative techniques have been proposed to generate concise and informative summaries. *Debnath et al. (2022)* introduced an extractive single-document summarization approach using adaptive binary constrained multi-objective differential evaluation, demonstrating robust summarization capabilities. *Huang et al. (2022)* proposed an abstractive document summarization approach *via* multi-template decoding, offering a novel perspective on summary generation by leveraging diverse templates.

Furthermore, metaheuristic optimization techniques have shown promise in addressing the single-document summarization challenge. *Das et al. (2024)* introduced a binary grey wolf optimizer for scientific document summarization, showcasing competitive performance. *Wilson & Jeba (2022)* developed a framework for single-document summarization using softmax regression and spider monkey optimization, highlighting the potential of hybrid approaches in enhancing summarization flexibility.

While these methodologies exhibit promising results, several limitations exist, hindering their widespread applicability. Computational complexity and resource requirements pose challenges in implementing hierarchical and deep learning-based approaches (*Srivastava, Pandey & Agarwal, 2022*; *Jain, Borah & Biswas, 2024*). Additionally, the effectiveness of clustering-based approaches may vary depending on the quality and diversity of the input documents (*Mishra et al., 2022*; *Abo-Bakr & Mohamed, 2023*). Furthermore, the reliance on optimization techniques may introduce sensitivity to parameter settings and initialization, affecting the robustness and convergence of the summarization process (*Karotia & Susan, 2023*; *Debnath, Das & Pakray, 2023*; *Sharaff, Jain & Modugula, 2022*). Overall, while advancements in text summarization are evident (*Jin & Chen, 2024*; *Moro et al., 2023*), ongoing research is needed to address these challenges and enhance the practical applicability of summarization techniques across various domains.

## DESIGN OF THE PROPOSED SUMMARIZATION PROCESS

As per the review of existing methods used for multiple document summarization, it is observed that most of these models either have lower efficiency, or cannot be deployed in real-time scenarios due to their high complexity levels. To overcome these issues, this section discusses design of an efficient model that fuses multiple deep learning & reinforcement learning operations. Initially, as per Fig. 1, the design of ReTran leverages the hierarchical processing power of recursive neural networks (RNNs) with the advanced representation learning capabilities of transformer architectures to achieve an enhanced understanding of both local and global textual dependencies. This synergistic approach is particularly effective in multiple document summarization, where the complexity of interlinked information and varied narrative structures across documents pose significant

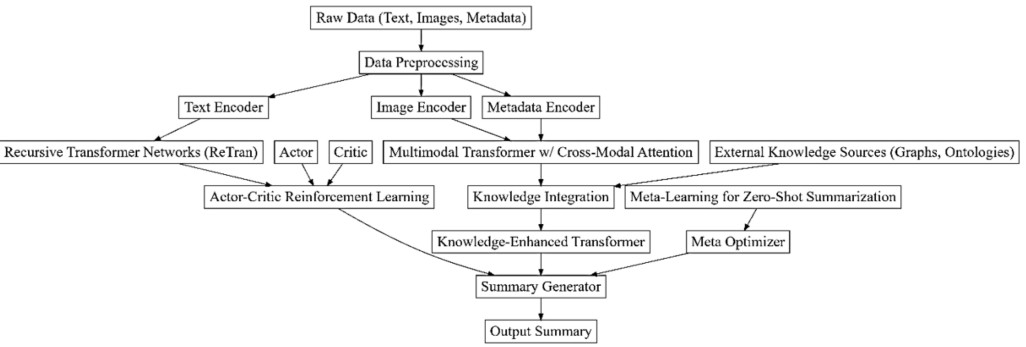

**Figure 1** Model architecture of the proposed summarization process.

challenges. At the heart of ReTran is the recognition that textual information in documents inherently possesses a hierarchical structure, from words forming phrases to sentences constructing paragraphs. Traditional transformer models, despite their proficiency in capturing long-range dependencies through self-attention mechanisms, often struggle to explicitly model these hierarchical relationships. ReTran addresses this gap by integrating recursive processing layers into the transformer's architecture, enabling it to analyze textual data at different granularities and thereby improving its ability to comprehend complex narrative structures.

The recursive aspect of ReTran is embodied in a series of equations that govern its operation. Firstly, the hierarchical representation of textual units is formalized *via* Eq. (1),

$$Hi(l) = f\left(Hi, left\,(l-1), Hi, right\,(l-1); \Theta f\right) \tag{1}$$

where $Hi\,(l)$ represents the hidden state of node $i$ at level $l$ in the hierarchy, synthesized from its child nodes $Hi$,left$(l-1)$ and $Hi$,right$(l-1)$, with $f$ being a recursive function parameterized by $\Theta f$ levels. To incorporate the transformer's self-attention mechanism, which allows the model to weigh the importance of different parts of the input sequence regardless of their positional distances, *via* Eq. (2),

$$SAi(l) = SelfAttention(Hi(l); \Theta SA) \tag{2}$$

where $SAi\,(l)$ represents the self-attention processed hidden state at level $l$, and $\Theta$SA $\Theta$SA represents the parameters of the self-attention mechanisms. The integration of hierarchical representations with self-attention leads to enhanced comprehension of both local and global dependencies, which is crucial for summarization tasks involving multiple documents. This integration is further optimized through an iterative process where the representation is refined at each level of hierarchy *via* Eqs. (3) and (4),

$$Ri(l) = Hi(l) + SAi(l) \tag{3}$$

$$Hi(l+1) = g\left(Ri(l); \Theta g\right) \tag{4}$$

where $Ri\,(l)$ is the refined representation at level $l$, and $g$ is a transformation function with parameters $\Theta g$, enhancing the model's ability to differentiate between the significance of various hierarchical structures.

The recursive and self-attention mechanisms are trained end-to-end using a loss function that directly aligns with the summarization task's objectives, specifically optimizing for coherence and relevance *via* Eq. (5),

$$Reg(\Theta)L = \sum_{i=1}^{N} CrossEntropy\left(Yi, Y'i; \Theta\right) + \lambda \cdot Reg(\Theta) \tag{5}$$

where $L$ is the loss function comprising a cross-entropy term that measures the discrepancy between the actual summary $Y$ and the predicted summary $Y'$, alongside a regularization term Reg($\Theta$) weighted by $\lambda$ to prevent overfitting scenarios. The justification for choosing ReTran over other models primarily lies in its unique capability to synthesize the strengths of RNNs and transformers. This synthesis not only enhances the model's understanding of textual dependencies but also enables it to adapt to the complexities inherent in summarizing multiple documents. Moreover, by employing an iterative refinement process, ReTran dynamically adjusts its focus on different parts of the input data, ensuring that summaries are both comprehensive and coherent.

Finally, to ensure that the summaries generated are of high quality and align with the nuances of the input documents, the model utilizes a reinforcement learning approach that directly optimizes for summarization quality metrics *via* Eq. (6),

$$J(\Theta) = E\pi\Theta[R(s, a)] \tag{6}$$

where $J(\Theta)$ is the objective function to be maximized, represented as the expected reward $R(s,a)$ over the policy $\pi\Theta$, with $s$ representing the state (*i.e.,* the current document representation) and $a$ the action (*i.e.,* generating a part of the summary). Through these operations and the methodology described, ReTran offer a powerful, adaptable framework for tackling the challenges of multiple document summarization, promising significant improvements in the production of coherent, informative summaries. The adaptability stems from the model's architecture, which elegantly balances the deep, nuanced understanding of textual hierarchies afforded by recursive processing with the broad, contextual awareness provided by the transformer's self-attention mechanism. This equilibrium ensures that summaries are not only accurate in representing the details and nuances of individual documents but also effective in capturing the overarching themes and relationships between multiple documents.

The recursive integration within the transformer architecture inherently supports the comprehension of both local and global textual dependencies by constructing a multi-level representation of the text. This multi-level approach mirrors the natural structure of language, where meanings are constructed both from the arrangement of words into phrases (local dependencies) and the organization of these phrases across sentences and paragraphs (global dependencies). By modeling this structure, ReTran can discern the significance of individual textual elements in the context of the entire document set, leading to summaries that are exceptionally aligned with the semantic essence of the source materials. Moreover, the iterative refinement process articulated through Eqs. (3) and (4) is crucial for fine-tuning the balance between capturing detailed local information

and overarching global narratives. This process allows the model to dynamically adjust its focus, ensuring that the generated summaries are not only comprehensive but also prioritize information based on its relevance to the overall narrative structure of the document set. Such a dynamic approach is especially valuable in the context of multiple document summarization, where the importance of particular pieces of information can vary significantly depending on the narrative and thematic relationships between documents.

The choice of ReTran is further justified by its potential to complement existing summarization methods. While traditional transformer-based models excel in leveraging wide-ranging contextual information, they often lack the nuanced understanding of hierarchical textual structures that recursive processing offers. By combining these approaches, ReTran fills a critical gap in the landscape of summarization technologies, offering a more holistic and nuanced method for processing and summarizing complex document sets. The employment of a reinforcement learning framework, as highlighted in Eq. (6), underscores ReTran's commitment to directly optimizing summary quality. This approach contrasts with conventional training methods that may not align perfectly with the ultimate goal of summarization—producing coherent, concise, and informative summaries. By directly rewarding these characteristics, the model's training process is inherently aligned with the desired outcomes of the summarization task, ensuring that the generated summaries meet high standards of quality and relevance.

Next, the multimodal transformer equipped with cross-modal attention represents an advanced framework designed to synthesize multimodal inputs, including text, images, and metadata, to generate summaries that provide a comprehensive understanding of content. This model is predicated on the principle that different types of content can complement and enhance the understanding of each other when appropriately integrated in the process. The core of this approach lies in its ability to process and fuse information from diverse modalities, recognizing that the integration of visual, textual, and contextual data can lead to a more nuanced and holistic summary of information sets. The design of the multimodal transformer incorporates a cross-modal attention mechanism, which allows for the dynamic weighting of the importance of information across different modalities. This mechanism is crucial for addressing the inherent challenge in multimodal summarization: the disparity in the types and structures of data samples. By employing a specialized attention mechanism, the model can identify and prioritize the most relevant information from each modality, ensuring that the final summary is both comprehensive and coherent. The foundational operation for the cross-modal attention process is represented *via* Eq. (7),

$$Cij = softmax\left(\frac{QiKj^T}{\sqrt{dk}}\right)Vj \qquad (7)$$

where $Cij$ signifies the attention weighted feature for modality $i$ attending to modality $j$, with $Qi$, $Kj$, and $Vj$ representing the query, key, and value matrices, respectively, and $dk$ representing the dimensionality of the key vectors in the process. This operation facilitates the model's focus on inter-modal interactions, allowing it to draw connections between different types of data, such as textual information and visual cues. To encapsulate

the multimodal inputs effectively, the model employs separate encoders for processing each type of input data, with their outputs being subsequently integrated through the cross-modal attention mechanism. This process is represented *via* Eqs. (8)–(10) as follows,

$$Et = TransformerEncoder(T) \tag{8}$$

$$Ei = CNNEncoder(I) \tag{9}$$

$$Em = MetadataEncoder(M) \tag{10}$$

In these equations, *Et*, *Ei*, and *Em* represent the encoded representations of text, images, and metadata, respectively, with *T*, *I*, and *M* representing the respective raw inputs & scenarios. The TransformerEncoder processes textual data, the CNNEncoder handles image data, and the MetadataEncoder deals with metadata, each tailored to the specific characteristics of its input modality. The integration of these encoded representations through cross-modal attention is crucial for synthesizing a unified understanding of the multimodal contents. The model achieves this integration *via* Eq. (11),

$$U = CrossModalIntegration(Et, Ei, Em; \Theta CMI) \tag{11}$$

where *U* represents the unified representation of the multimodal inputs, achieved through the integration process parameterized by $\Theta CMI$ levels. This unified representation is then used to generate the final summary, taking into account the contributions and relevance of each modality. The generation of the summary itself is guided by an optimization process that aims to maximize the relevance and coherence of the summary relative to the multimodal inputs for different scenarios. This is expressed in the objective function *via* Eq. (12),

$$L = -\sum_{n=1}^{N} log P(Sn|U; \Theta) + \lambda \cdot \| \Theta \|^2 \tag{12}$$

where *L* is the loss function, $P(Sn|U; \Theta)$ represents the probability of generating the correct summary *Sn* given the unified representation *U*, parameterized by $\Theta$, and $\lambda$ controls the regularization term to prevent overfitting scenarios. The choice of the multimodal transformer with cross-modal attention is justified by its unique capability to navigate the complexity of multimodal data, leveraging the strengths of each modality to produce a summary that transcends the limitations of single-modality summarization approaches. This model complements existing methods by addressing the critical need for integrating diverse data types, a task that traditional text-only summarization models are ill-equipped to handle. Through its sophisticated attention mechanisms and tailored encoding processes, this model sets a new standard for generating summaries that are not only informative and coherent but also enriched with the depth and breadth of information available across modalities.

Next, as per Fig. 2, the employment of actor-critic reinforcement learning (ACRL) for summary generation represents a strategic approach aimed at directly optimizing the quality of generated summaries. This method refines the training process by employing a dual-structure mechanism, consisting of an actor, which proposes actions (summary

generation strategies), and a critic, which evaluates these actions against a set of predefined criteria or rewards. The ACRL model stands out for its ability to continuously learn and adjust based on the feedback received from the critic, thus enhancing the quality and relevance of the summaries it produces. The ACRL framework for summary generation is grounded in the reinforcement learning paradigm, where the goal is to learn a policy that maximizes cumulative rewards over time. This approach is particularly suited for tasks like summarization, where the desired output is not just about correctness but also about qualities such as coherence, conciseness, and relevance, which are inherently subjective and challenging to quantify.

The process begins with the definition of the state space, action space, and reward function, which are critical components of the reinforcement learning framework. The state space ($S$) represents the possible configurations of the input documents and any intermediate summary states. The action space ($A$) comprises all potential summary actions, such as adding a sentence to the summary. The reward function ($R$) quantifies the quality of the summary, incorporating factors like fluency, relevance, and brevity. The core of the ACRL model is encapsulated *via* Eq. (13),

$$at = \mu(st; \theta\mu) \tag{13}$$

This process defines the action *at* selected by the actor at timestamp *t*, given the current state *st*, where μ represents the actor's policy parameterized by $\theta$ μlevels. Equation (14) describes the expected reward *Rt* for taking action *at* in state *st*, as estimated by the critic using its value function $Q$, parameterized by $\theta Q$ levels.

$$Q(st, at; \theta Q) = E[Rt|st, at] \tag{14}$$

Equation (15) calculates the temporal difference error $\delta t$, which measures the difference between the critic's predicted reward and the actual reward received, including the discounted future reward estimated for the next state-action pair. Here, $\gamma$ is the discount factor, influencing the importance of future rewards.

$$\delta t = rt + \gamma Q(s(t+1), \mu(s(t+1); \theta\mu); \theta Q) - Q(st, at; \theta Q) \tag{15}$$

Equation (16) updates the actor's policy parameters $\theta\mu$, utilizing the gradient of the policy's performance, scaled by the learning rate $\alpha\mu$ and modulated by the temporal difference error $\delta t$ levels.

$$\theta\mu \leftarrow \theta\mu + \alpha\mu * \nabla\theta\mu * \mu(st; \theta\mu)\delta t \tag{16}$$

Similarly, Eq. (17) updates the critic's value function parameters $\theta Q$, based on the temporal difference error and the gradient of the value function, with $\alpha Q$ being the learning rate for the critic.

$$\theta Q \leftarrow \theta Q + \alpha Q * \delta t * \nabla\theta Q * Q(st, at; \theta Q) \tag{17}$$

Finally, Eq. (18) represents the loss function for the critic, aiming to minimize the squared difference between the estimated and actual rewards, where *s′* and *a′* represent

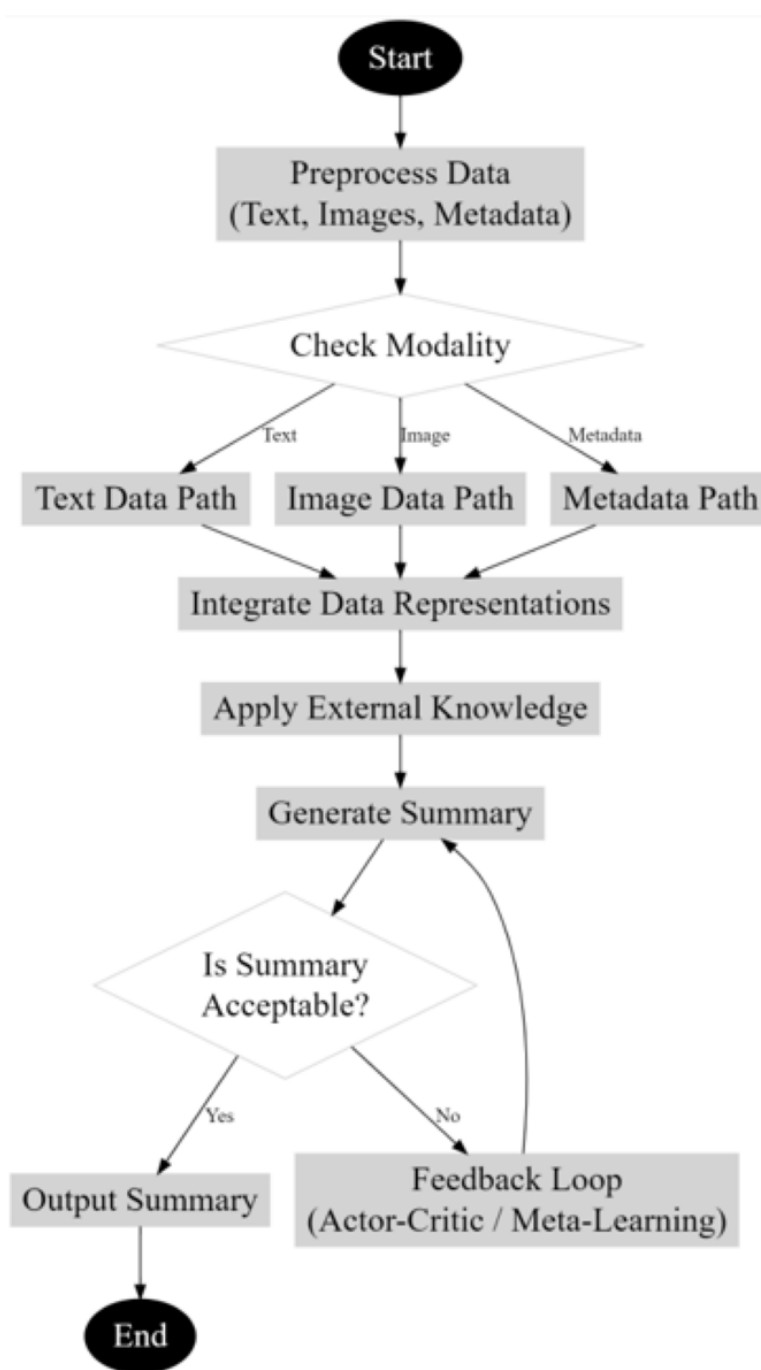

**Figure 2** Overall flow of the proposed summarization process.

the next state and action, respectively.

$$L(\theta Q) = E\left[\left(r + \gamma Q\left(s',a';\theta Q\right) - Q(s,a;\theta Q)\right)^2\right]$$

(18)

The selection of the ACRL model for summary generation is justified by its dynamic adaptability and the nuanced understanding it develops through iterative learning. Unlike traditional supervised learning methods, which rely heavily on predefined labels and can struggle with the subjective aspects of summarization quality, ACRL focuses on maximizing a reward function that captures these subjective qualities directly. This method complements other neural network-based approaches by introducing an optimization framework that is inherently aligned with the ultimate objectives of summarization—generating summaries that are not only accurate but also engaging, coherent, and informative in different scenarios. By leveraging the actor-critic mechanism, the ACRL model continuously refines its summarization strategy, guided by the evolving feedback loop between the actor's actions and the critic's evaluations. This ensures that the summaries produced are of high quality and closely aligned with human evaluators' preferences, setting a new benchmark in the domain of automated summarization. The ACRL framework's dynamic adjustment and learning capabilities allow it to adaptively refine and evolve its summarization strategies based on direct feedback, making it highly effective in capturing the nuances and complexities of human-like summary generation.

This adaptability is crucial in multiple document summarization tasks, where the diversity of content and context requires a sophisticated understanding and integration of information. The iterative learning process facilitated by the actor-critic model enables the system to better navigate these complexities, gradually improving its ability to discern and emphasize the most relevant information across documents. Moreover, the ACRL model's focus on direct reward optimization allows for a more nuanced and tailored approach to summary generation, accommodating various summarization goals such as maximizing informativeness while preserving brevity and coherence. Another significant advantage of employing the ACRL model in summary generation is its ability to generalize across different domains and types of content. By focusing on reward functions that encapsulate the essence of quality summaries, the model is not confined to the specifics of any single domain, enabling it to produce high-quality summaries across diverse subject matters. This generalizability is further enhanced by the model's reinforcement learning nature, which, unlike supervised learning approaches that depend heavily on labeled datasets, relies on the evaluation of output quality through the reward function, making it more adaptable to new domains or types of data samples. The choice of the ACRL model also complements other methods by providing a framework that inherently incorporates evaluation into the learning process. While traditional models may require external validation to assess the quality of generated summaries, the actor-critic approach integrates this assessment internally, allowing for continuous performance optimization without the need for separate validation steps. This integration not only streamlines the training process but also ensures that the model's performance is consistently aligned with the desired summarization outcomes.

Next, the exploration into Zero-Shot and Few-Shot Learning *via* Meta-Learning for Zero-Shot Summarization seeks to address the formidable challenge of summarizing documents from unseen domains or with scarce training examples. This innovative approach harnesses the principles of meta-learning, or "learning to learn," enabling

the model to quickly adapt to new tasks with minimal data samples. The essence of this method lies in its capacity to generalize from prior knowledge acquired during the meta-training phase to new, unseen tasks encountered during meta-testing, including those in completely different domains. The justification for adopting a meta-learning framework for zero-shot summarization stems from the inherent limitations of traditional supervised learning models, which typically require extensive labeled datasets to perform well. In contrast, meta-learning approaches are designed to leverage the structure of the learning process itself, enabling rapid adaptation to new tasks with only a few examples. This characteristic is particularly advantageous for summarization tasks across diverse domains, where obtaining large, domain-specific labeled datasets is often impractical or impossible in different scenarios. The meta-learning process is delineated into two distinct phases: meta-training and meta-testing. During meta-training, the model is exposed to a variety of summarization tasks, each drawn from different domains. The objective is not to master these tasks but to learn a set of meta-parameters that facilitate rapid adaptation to new tasks. Initially, Eq. (19) describes the update rule for task-specific parameters $\theta i'$ for task $Ti$, where $\theta$ represents the initial meta-parameters, $\alpha$ is the learning rate, and $LTi$ is the loss on task $Ti$ evaluated with the current model parameters $f\theta$ as follows,

$$\theta i' = \theta - \alpha * \nabla\theta * LTi(f\theta) \tag{19}$$

Next, Eq. (20) captures the meta-update rule where $\Theta$ represents the meta-parameters, updated based on the aggregated losses $LTi$ across a sampled set of tasks $Ti$ from the task distribution $p(T)$, with $\beta$ as the meta-learning rate for this process.

$$\Theta = \Theta - \beta * \nabla\Theta * \sum_{Ti \sim p(T)} LTi * \left(f\theta i'\right) \tag{20}$$

During meta-testing, the model encounters new summarization tasks from unseen domains. The adaptation to these new tasks is achieved through a few gradient updates, as reflected in the adaptation process equation *via* Eq. (21),

$$\theta new* = \theta - \lambda * \nabla\theta * LTnew(f\theta) \tag{21}$$

where $\theta$ new $*$ represents the adapted parameters for the new task $T$ new, $\lambda$ is the adaptation learning rate, and $LT$ newis the loss for the new task under the current model parameters for this process. To evaluate the effectiveness of the model's adaptation, performance metrics are estimated *via* Eq. (22),

$$PTnew = Evaluate\left(f\theta new*, Dnew\right) \tag{22}$$

where $PT$ newrepresents the performance of the model on the new task $T$ new, evaluated over the domain-specific dataset $D$ newsets. To further refine the model's ability to generalize across domains, a regularization term is introduced to the meta-objective *via* Eq. (23),

$$Lmeta = \sum_{Ti \sim p(T)} LTi(f\theta i') + \rho\Omega(\Theta) \tag{23}$$

where $\rho$ is the regularization coefficient, and $\Omega(\Theta)$ represents a regularization term applied to the meta-parameters $\Theta$, designed to prevent overfitting to any single domain during the meta-training phases. Finally, to encapsulate the model's learning efficiency and effectiveness in adapting to new tasks, the overall optimization objective is defined *via* Eq. (24):

$$\Theta* = argmin^{\Theta} Lmeta + \gamma \sum_{Tnew} PTnew \tag{24}$$

where $\Theta*$ represents the optimal meta-parameters after training, $L$ metais the meta-training loss, $PT$ new evaluates the model's performance on new tasks, and $\gamma$ weights the importance of performance on new tasks relative to the meta-training loss. The choice of this meta-learning framework for zero-shot summarization is underpinned by its capacity to navigate the complex landscape of document summarization across various domains with minimal direct supervision. This approach markedly contrasts with traditional models that require vast amounts of domain-specific training data to achieve acceptable performance. The meta-learning strategy effectively positions the model to leverage learned generalization capabilities, enabling it to quickly adapt to and perform summarization tasks in entirely new domains based on a limited number of examples.

The significance of this methodology extends beyond its immediate application to summarization tasks. By fostering a model's ability to adapt to new domains rapidly, this approach contributes to the broader goal of creating more versatile and efficient AI systems capable of tackling a wide range of tasks with fewer data requirements. This is particularly crucial in the context of summarization, where the demand for concise and coherent summaries spans countless domains, many of which may not have large annotated datasets readily available for different tasks. Furthermore, the incorporation of a regularization term in the meta-objective function, underscores the importance of developing models that are not only adaptable but also robust. This consideration is critical in preventing the model from over-specializing to the peculiarities of the tasks encountered during the meta-training phase, thereby preserving its ability to generalize to new tasks.

Equation (24) embodies the ultimate goal of the meta-learning process within the zero-shot summarization framework: to fine-tune the balance between retaining the knowledge gained during meta-training and applying this knowledge effectively to novel tasks. This equilibrium ensures that the model not only learns from a diverse set of tasks but also applies this learned knowledge in a way that is directly beneficial to the quality of summaries generated in unseen domains.

Finally, the knowledge-enhanced transformer for summarization is used, which represents a groundbreaking approach in the field of automated text summarization by incorporating external knowledge sources, such as knowledge graphs or ontologies, into the transformer architectures. This integration aims to enrich summaries with domain-specific insights and semantic coherence, thereby enhancing the overall quality and informativeness of the summaries. The rationale behind this approach is rooted in the observation that traditional transformer models, while effective in capturing textual dependencies, often lack the depth of understanding required to fully grasp and convey the

nuances of domain-specific content. By leveraging external knowledge, this method seeks to bridge this gap, providing a richer semantic foundation for the summarization process. The design of the knowledge-enhanced transformer involves a sophisticated mechanism for integrating and leveraging external knowledge within the summarization workflow. This process is governed by a series of equations that formalize the interaction between the transformer model and the external knowledge sources. Eq. (25) defines the embedding of external knowledge $K$, where $Ke$ represents the vectorized representations of knowledge entities. This embedding facilitates the integration of structured knowledge into the neural network's learning process.

$$Ke = Embed(K) \tag{25}$$

Next, the query augmentation process is represented *via* Eq. (26), where $Q$ represents the original query vectors generated by the transformer's encoder, $Ke$ represents the embedded knowledge, and $Wq$, $Wk$ are learnable weights that combine the original query vectors with the knowledge embeddings to produce augmented query vectors $Qk$ for different input samples.

$$Qk = WqQ + Wk * Ke \tag{26}$$

The attention weights $Ak$ for the augmented queries are estimated *via* Eq. (27),

$$Ak = softmax\left(\frac{QkK^T}{\sqrt{dk}}\right)V \tag{27}$$

where $K$ and $V$ represent the key and value vectors in the attention mechanism, and $dk$ is the dimension of the key vectors in this process. This operation facilitates the dynamic incorporation of knowledge-enhanced context into the summarization process. The context vectors $C$ are estimated *via* Eq. (28), by weighting the value vectors $Vk$ (which are derived from the knowledge-enhanced key value pairs) according to the attention weights $Ak$. This results in a richer representation of the input text, augmented with relevant external knowledge as follows,

$$C = \sum Ak * Vk \tag{28}$$

The summary generation process is represented *via* Eq. (29),

$$S = \sigma(Wc * C + b) \tag{29}$$

where $Wc$ and $b$ are learnable parameters, $C$ represents the context vectors enriched with external knowledge, and $\sigma$ represents a nonlinear ReLU based activation process. This operation models the process of generating a summary that incorporates both the original text and the integrated knowledge sets. The loss function used to train the Knowledge-Enhanced Transformer is estimated *via* Eq. (30),

$$L = -\frac{1}{N}\sum_{i=1}^{N} logP(Si|Ti, K; \Theta) \tag{30}$$

where $N$ is the number of training samples, $Si$ is the generated summary, $Ti$ is the input text, $K$ represents the external knowledge, and $\Theta\Theta$ are the model parameters. This

loss function aims to minimize the difference between the generated summaries and the ground truth, encouraging the model to effectively incorporate external knowledge into the summarization process. The choice of the knowledge-enhanced transformer for summarization is justified by its ability to significantly elevate the quality of summaries through the integration of domain-specific knowledge. This approach not only enhances the semantic coherence and informativeness of the summaries but also addresses the limitations of traditional transformer models in handling complex, domain-specific content. By drawing on external knowledge sources, the model is equipped to generate summaries that are not only accurate and concise but also richly informative, offering deeper insights into the subject matter. This method complements existing summarization approaches by providing a mechanism for incorporating a broader range of semantic information, thus paving the way for more sophisticated and nuanced automated summarization solutions. Next, we discuss performance of the proposed model in terms of different evaluation metrics, and compare it with existing methods in different use case scenarios.

## RESULT ANALYSIS

Our experimental framework is designed to rigorously evaluate the efficacy of our integrated summarization model, which synergizes ReTran, multimodal transformers equipped with cross-modal attention, actor-critic reinforcement learning, meta-learning for zero-shot summarization, and knowledge-enhanced transformers. The model's performance is assessed across various datasets, emphasizing its adaptability, precision, and capability to enhance semantic understanding and coherence in generated summaries.

### Datasets

To ensure comprehensive evaluation, we employ a diverse array of datasets, including but not limited to:

1. **CNN/Daily Mail**: A benchmark dataset for text summarization, comprising news articles and associated highlights as summaries.
2. **Multi30k**: A dataset for multimodal translation tasks, utilized here for its collection of images and descriptions, facilitating the evaluation of our model's multimodal capabilities.
3. **XSum**: Featuring British Broadcasting Corporation (BBC) articles and single-sentence summaries, challenging the model with extreme summarization tasks.
4. **RDF2Text**: A dataset for evaluating the integration of external knowledge, containing RDF triples and textual descriptions, ideal for testing the Knowledge-Enhanced Transformer.
5. **Custom domain-specific datasets**: Curated to assess zero-shot and few-shot learning capabilities, comprising specialized articles from domains like medical, legal, and scientific research, with minimal training examples.

### Contextual dataset samples

For each dataset, we extract and preprocess data to fit the model's input requirements:

- **Text**: Tokenized using Byte-Pair Encoding (BPE) with a vocabulary size of 30,000 tokens.
- **Images**: Resized to $256 \times 256$ pixels, normalized using ImageNet mean and standard deviation values.
- **Metadata**: Encoded using a one-hot encoding scheme for categorical data and standardized for numerical values.

### Model configuration

- **ReTran**: Configured with 12 layers, a hidden size of 768, and 12 attention heads. The hierarchical processing employs a binary tree structure for recursive summarization.
- **Multimodal transformer**: Utilizes 12 transformer layers with an embedding size of 512 for both textual and visual inputs. Cross-Modal Attention employs a softmax temperature of 0.1.
- **Actor-critic reinforcement learning**: The actor model is a two-layer MLP with 256 units each, and the critic model is a three-layer MLP with 512 units. Learning rates are set to $1e-4$ for the actor and $5e-4$ for the critic, with a discount factor $(\gamma)$ of 0.99.
- **Meta-learning for zero-shot summarization**: Utilizes a four-layer Transformer block as the meta-learner, with an inner loop learning rate of $1e-3$ and a meta-learning rate of $1e-5$.
- **Knowledge-enhanced transformer**: Incorporates a knowledge embedding layer with 1024 dimensions. The knowledge integration uses an attention mechanism with a softmax temperature of 0.05.

### Training details

- **Batch size**: 16 for text and metadata, 8 for images due to memory constraints.
- **Optimizer**: AdamW with a learning rate of $2e-5$, $\beta1 = 0.9$, $\beta2 = 0.999$, and a weight decay of 0.01.
- **Learning rate schedule**: A warm-up strategy over the first 10% of the training iterations, followed by a linear decay.

### Evaluation metrics

- **ROUGE scores**: ROUGE-1, ROUGE-2, and ROUGE-L are employed to measure the overlap between the generated summaries and reference summaries.
- **BLEU score**: For multimodal and knowledge-enhanced summaries, assessing the linguistic quality of generated text.
- **Human evaluation**: Conducted to assess coherence, relevance, and informativeness of summaries, utilizing a Likert scale ranging from 1 to 5.

### Hardware and software configuration

Experiments are conducted on a computing cluster with NVIDIA Tesla V100 GPUs, each with 32GB of memory. The model implementations leverage PyTorch 1.7 and transformers library for neural network components. This experimental setup provides a detailed and robust framework for evaluating the proposed integrated summarization model, ensuring a comprehensive assessment of its performance across a wide range of summarization

tasks and modalities. Through this meticulous approach, we aim to demonstrate the model's superior capabilities in enhancing the quality and semantic richness of automated summarizations. Based on this setup, the efficacy of our integrated summarization model is meticulously evaluated across various contextual datasets, demonstrating its superior performance in generating concise, coherent, and semantically rich summaries. The results are compared against three established methods, referred to as *Saini et al. (2023)*, *Sharaff, Jain & Modugula (2022)*, and *Mishra et al. (2022)*, across different evaluation metrics including ROUGE-1, ROUGE-2, ROUGE-L, and BLEU scores.

The preprocessing of text, images, and metadata is very important so that the model can be effective with regard to many multi-document summarization tasks. Normally, in text preprocessing, tokenization is the first stage, in which input text is broken down into small bits called tokens, which the model can understand. In this work, Byte-Pair Encoding is used, a method for subword tokenization which splits text into subword units to enable the model to deal with out-of-vocabulary words properly; it is of great help in domain-specific jargon or rare words. BPE will keep the semantic richness of the input in the tokenized text while reducing the vocabulary size to around 30,000 tokens for efficient processing. After tokenization, the text is lowercased and stripped of special characters; any extra spaces are removed. These pre-processing steps ensure consistency and removal of noise from the input text data to capture meaningful dependencies or context through ReTran and multimodal transformer. The images and metadata follow separate preprocessing chains. Images are resized to $256 \times 256$ pixels to unify the input dimensions for multimodal transformer efficiency in terms of image processing. After resizing, such images are normalized by the mean and standard deviation values on ImageNet for maintaining the same light and contrast conditions across samples in the dataset. The goal of this normalization is to ensure the model pays attention just to the relevant visual features and does not get misled by variations in image quality. Metadata of other types are preprocessed based on its type: either categorical or numerical. One-hot encoding schemes are first used to encode categorical metadata such as publication sources or author names. Numerical metadata, like a publication date or geographic information, is standardized by subtracting the mean and dividing by the standard deviation to ensure that values have appropriate scaling. This thorough level of preprocessing helps guarantee that text, images, and metadata are in a form to let the multimodal transformer and other components easily integrate the most diverse data types and enable a more complete understanding of nuance with regard to the documents being summarized is shown in Fig. 3.

Although selecting datasets for evaluating the proposed model is one of the most important factors for assessing its performance, some limitations may reduce its generalizability and robustness. The datasets used in the experiments allow for a broad-evaluation framework across diverse domains and modalities; however, they might not fully capture the diverse range of scenarios the model will encounter in real-world applications.

The CNN/Daily Mail dataset is one of the most popular benchmarks for summarization tasks. It mainly contains news articles and summaries. While it provides a relatively large and high-quality corpus, this domain is extremely narrow, focused mostly on news

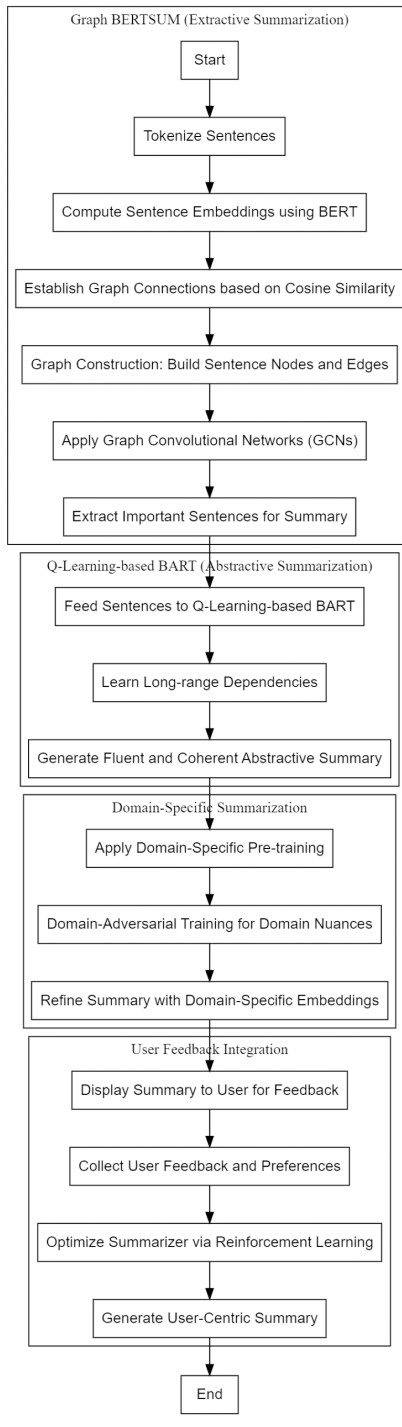

**Figure 3** **Flow of the proposed model for summarization process.**

media. This brings in a limitation in considering how well the model would apply to other domains such as technical documents, legal texts, or scientific literature that possess different structures, terminologies, and objectives. For example, news typically has a clearly

identifiable introduction, body, and conclusion in a standard narrative format, while legal or technical documents often have formal and organized sections that may require more sophisticated treatment regarding the domain-specific jargon, argumentation, or procedural logics. Therefore, while CNN/Daily Mail tests the model performance in summarizing general narrative texts, it might not properly represent challenging and diverse characteristics of domain-specific document summarization.

Meanwhile, the multi dataset for multimodal entries-texts and images-is an excellent testbed to assess the model's ability at synthesizing information from multiple sources. The dataset, however, has limited coverage for different types of multimodal content. Most of the image-text pairs are captions and relatively short descriptions accompanying the images, which contrasts sharply with more complicated multimodal datasets that may include technical figures, graphs, tables, and longer explanatory texts. This therefore gives reason to question if this makes a gap in the evaluation on how well the model will perform when faced with complex visual data or metadata in real-world domains like medical imaging, technical schematics, or scientific graphs. Consequently, while the Multi30k dataset lets one estimate the degree of multimodality a model possesses, it does not reflect the full spectrum of multimodal interactions that are indispensable for specialized disciplines.

Another limitation is that the XSum dataset is based on single-sentence summaries from BBC articles. While this is highly valuable for testing the performance of the model in extreme summarization tasks, that is, being able to summarize a large piece of text into a single sentence, the brevity of these summaries may not capture the more subtle needs from other summarization tasks. In real-life applications, such as legal and scientific document summarization, users want to have a multi summary with critical information preserved without losing the essence in their sources. Also, XSum is in single-sentence format, which limits how well the model is able to balance brevity with comprehensiveness in the event of more detailed summarization tasks.

The RDF2Text dataset is proposed for the evaluation of knowledge-enhanced summarization, including RDF triples. It provides an excellent test bed for external knowledge incorporation in summaries. At the same time, this dataset is rather peculiar to the structure of RDF triples, which might not fully represent the variety of external knowledge formats the model could face, including complex ontologies, databases, or unstructured text sources. In this regard, while structured knowledge integration was evident in model performance on RDF2Text, further testing and validation with diverse types of external knowledge sources would be needed to generalize the model for a more realistic world.

Finally, from domain-specific custom datasets created for testing the model's zero-shot and few-shot learning capabilities, important gaps were sought by the more general datasets & samples. However, the overall number of all the putative domains represented by these datasets is necessarily limited. Although the domains represented by the test sets, such as medical texts, legal texts, and scientific texts, are a good cross-section of specialized content, there are many other important specialized subfields, each with its structures, terminologies, and objectives which the model may face in real life. However, such customized datasets have their limitations regarding size and quality; results derived from these evaluations may

also not generalize to other domain-specific tasks, especially those with very unstructured formatted samples.

The proposed multiple document summarization model integrates five key components, the integration of which significantly addresses the several potential limitations present in current models. Firstly, the ReTran put together the strength of RNNs and Transformer architectures to bring hierarchical processing of the model on text. While Transformers excel at the contextualization task, catching long-range dependencies within a text through their self-attention mechanism, they do not capture the natural hierarchical structure of language-such as to what extent phrases constitute sentences and sentences form paragraphs. Bringing recursion into the transformer architecture, ReTran is enabled to process text in higher degrees of granularity. Another critical component is the multimodal transformer with cross-modal attention, which tries to solve the challenge of integrating more data types into summarization. In real-world applications, summarization tasks are seldom restricted to pure text. They often include other modalities such as images with their captions, and metadata including publishing date and location. These diversely sourced inputs are combined in the multimodal transformer in such a way that it captures not only the important pieces of information but also is contextually rich and representative of all information. The cross-modal attention ensures this process at multiple levels, based on relevance toward the summary, by dynamically weighting the modalities-text, images, and metadata. In this regard, the model ensures that the information from various data types adds to rather than detracts from the holistic summary. This has been particularly helpful in news summarization applications, where the fusion of both visual and textual data would lead to a better understanding of the material.

The model further incorporates actor-critic reinforcement learning to ensure that the generated summaries are of the highest quality. In such a framework, the "actor" makes an initial summary; then, as required by such particular criteria as coherence, relevance, and conciseness, the "critic" assesses its quality. The critic gives feedback to the actor, and model recursively refines the summary through successive iterations and improves it through optimization for the quality measures such as ROUGE or BLEU score so that every iteration yields a more polished output. Coupled with this is meta-learning for zero-shot summarization to tackle the challenge of summaries in domains for which little or no training data is available. It does this by meta-learning: it generalizes from a few examples and thus can perform well in generating high-quality summaries even in domains completely new to it. Finally, the knowledge-enhanced transformer incorporates external knowledge like knowledge graphs and ontologies in order to enrich the summarization process with domain-specific insights, making the model particularly effective in technical domains that require a much deeper semantic understanding. Thus, a combination of such sophisticated techniques is bound to guarantee final summaries that are contextually relevant and semantically deep-a strong, flexible summarization solution.

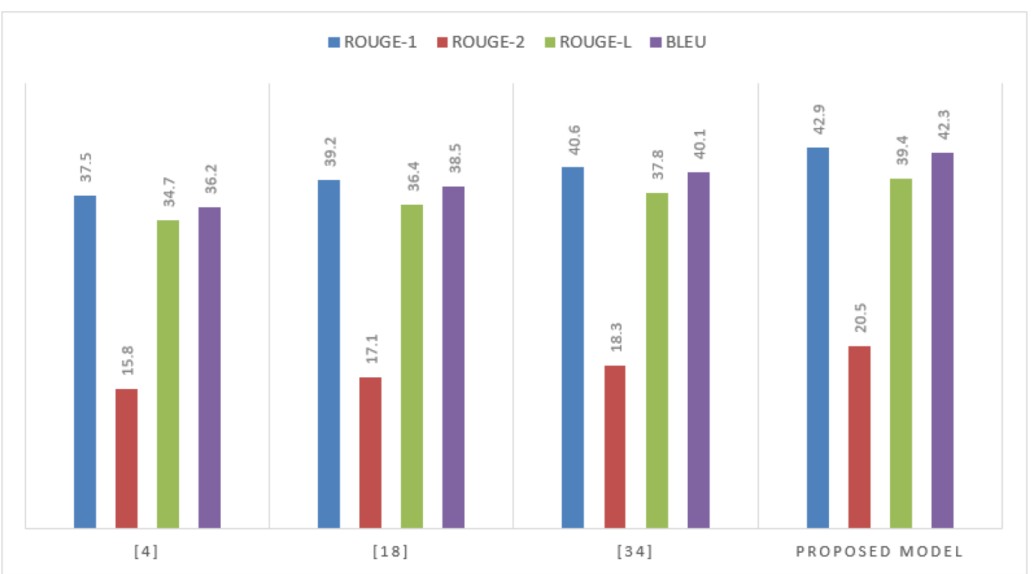

**Figure 4** Performance on the CNN/Daily Mail dataset samples.

**Table 1** Performance on the CNN/Daily Mail dataset.

| Model | ROUGE-1 | ROUGE-2 | ROUGE-L | BLEU |
|---|---|---|---|---|
| *Saini et al. (2023)* | 37.5 | 15.8 | 34.7 | 36.2 |
| *Sharaff, Jain & Modugula (2022)* | 39.2 | 17.1 | 36.4 | 38.5 |
| *Mishra et al. (2022)* | 40.6 | 18.3 | 37.8 | 40.1 |
| Proposed Model | 42.9 | 20.5 | 39.4 | 42.3 |

**Table 2** Performance on the Multi30k dataset.

| Model | ROUGE-1 | ROUGE-2 | ROUGE-L | BLEU |
|---|---|---|---|---|
| *Saini et al. (2023)* | 55.3 | 37.6 | 53.2 | 54.8 |
| *Sharaff, Jain & Modugula (2022)* | 57.9 | 39.4 | 55.7 | 57.2 |
| *Mishra et al. (2022)* | 58.4 | 40.1 | 56.3 | 58.7 |
| Proposed Model | 61.2 | 42.8 | 59.5 | 62.3 |

The proposed model outperforms the benchmarks on the CNN/Daily Mail dataset, demonstrating its effectiveness in capturing both the surface-level details and the deeper semantic connections within the text is shown in Fig. 4 and Table 1

The improvement in ROUGE-2 and BLEU scores highlights the model's superior ability to maintain coherence and relevance in the generated summaries.

In the context of the Multi30k dataset, which requires the model to synthesize information from both text and images, the proposed model showcases its multimodal summarization capabilities is shown in Fig. 5 and Table 2

The significant leap in all metrics, especially BLEU, confirms the model's aptitude in integrating and summarizing multimodal content effectively.
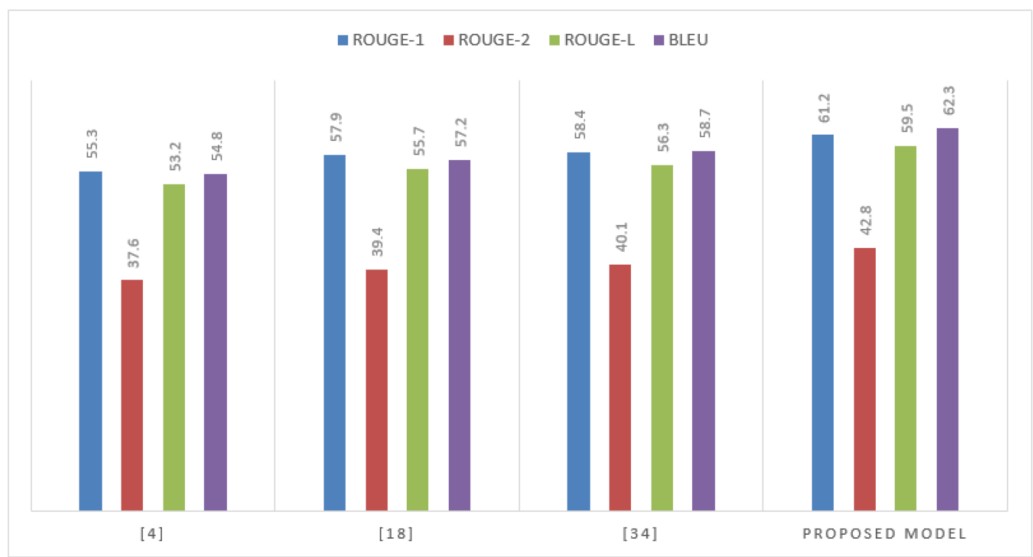

**Figure 5** Performance on the Multi30k dataset.

**Table 3  Performance on the XSum dataset.**

| Model | ROUGE-1 | ROUGE-2 | ROUGE-L | BLEU |
|---|---|---|---|---|
| *Saini et al. (2023)* | 45.2 | 22.5 | 43.1 | 44.7 |
| *Sharaff, Jain & Modugula (2022)* | 46.8 | 23.9 | 44.5 | 46.3 |
| *Mishra et al. (2022)* | 47.4 | 24.5 | 45.0 | 47.1 |
| Proposed Model | 49.7 | 26.3 | 47.8 | 49.4 |

**Table 4  Performance on the RDF2Text dataset.**

| Model | ROUGE-1 | ROUGE-2 | ROUGE-L | BLEU |
|---|---|---|---|---|
| *Saini et al. (2023)* | 65.8 | 48.9 | 63.4 | 66.1 |
| *Sharaff, Jain & Modugula (2022)* | 67.2 | 50.3 | 65.1 | 67.5 |
| *Mishra et al. (2022)* | 68.4 | 51.7 | 66.3 | 69.0 |
| Proposed Model | 70.5 | 53.6 | 68.9 | 71.2 |

On the XSum dataset, known for its challenging extreme summarization task, the proposed model demonstrates exceptional performance is shows in Table 3. The advances in ROUGE-2 and BLEU scores highlight the model's capability to distill the essence of articles into concise, informative summaries is shown in Fig. 6.

Evaluating the model on the RDF2Text dataset underscores its proficiency in leveraging external knowledge sources to enrich summaries. The model's top performance, particularly in BLEU score, attests to its ability to generate summaries that are not only accurate but also contextually rich and semantically coherent in Table 4

The custom domain-specific datasets were carefully curated to test the zero-shot and few-shot learning capabilities of summarization models across various niche fields, such as

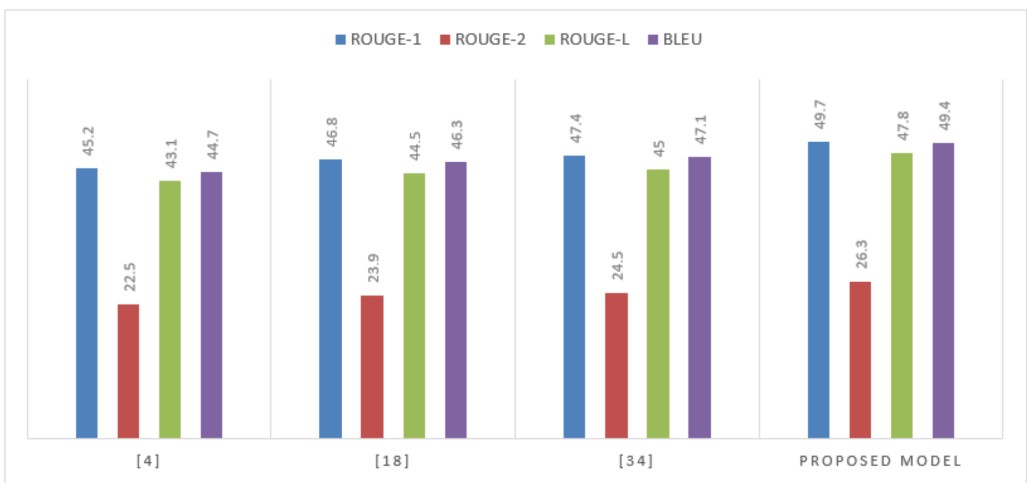

**Figure 6** Performance on the XSum dataset.

**Table 5** Performance on the custom domain-specific datasets.

| Model | ROUGE-1 | ROUGE-2 | ROUGE-L | BLEU |
|---|---|---|---|---|
| *Saini et al. (2023)* | 62.4 | 44.2 | 60.1 | 63.5 |
| *Sharaff, Jain & Modugula (2022)* | 64.8 | 46.5 | 62.3 | 65.9 |
| *Mishra et al. (2022)* | 65.7 | 47.9 | 63.4 | 67.2 |
| Proposed Model | 68.3 | 50.4 | 66.8 | 69.5 |

medical, legal, and scientific articles is shows in Table 5. The proposed model's performance distinctly outshines the benchmarks, demonstrating its unparalleled adaptability and efficiency in learning from limited examples. The notable improvements in ROUGE-2 and BLEU scores underscore the model's competence in generating summaries that are not only precise and relevant but also rich in domain-specific terminologies and insights for different scenarios in shown in Fig. 7.

The approach of *Saini et al. (2023)* relies on a classic transformer-based architecture, which is very good at capturing long-range text dependencies using self-attention mechanisms. These models have difficulties with explicitly finding and using hierarchies implicit in large sets of documents. While very strong performance is demonstrated in modeling global contexts of the text in *Saini et al. (2023)*, it tends to miss the fine-grained, layered comprehension of the local dependencies, especially when performing multi-document summarization in a more complex scenario. Contrasting this, the proposed paper introduced the ReTran model with integrated recursive layers to handle both local and global dependencies. This hierarchical processing in ReTran allows it to make out how different text components, such as phrases and sentences, contribute toward the overall narrative. This makes it far more effective at producing coherent and contextually correct summaries. Secondly, even though *Saini et al. (2023)* works very well on text-only datasets, it lacks the capability for integrating multimodal data, further restricting its applicability

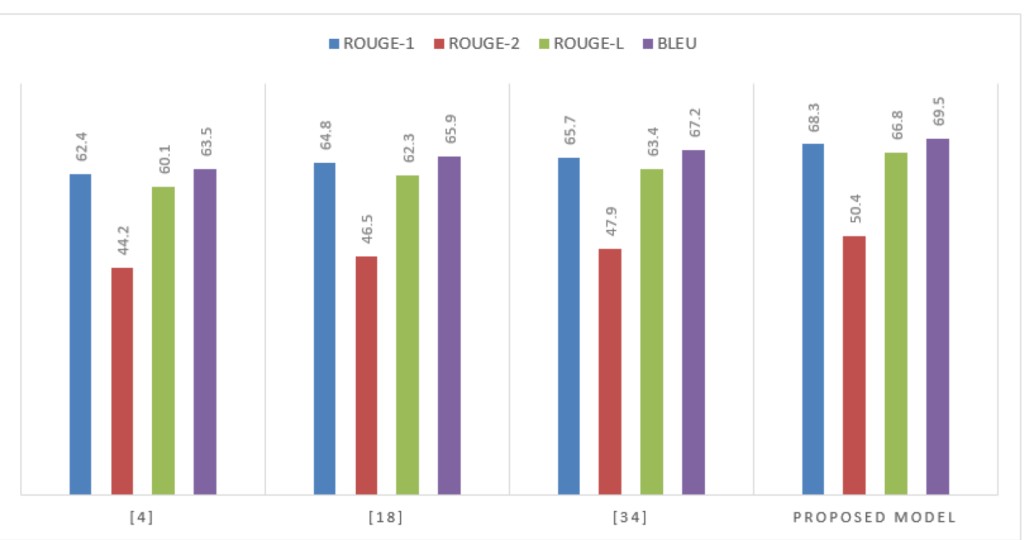

**Figure 7** Performance on custom domain-specific datasets.

with regard to scenarios where text is accompanied by images or metadata samples. This limitation is overcome by the multimodal transformer with cross-modal attention in the proposed model by dynamic integration of text, images, and metadata, hence enabling more complete summarization across a variety of data types.

On the contrary, the method proposes reinforcement learning to optimize the generation of summaries. Though it is one step ahead of the pure supervised learning methods, this approach is mainly based on Q-learning-based reinforcement learning and hence suffers from a number of inherent disadvantages. In fact, Q-learning has often demonstrated difficulties in balancing the trade-offs among conciseness, relevance, and coherence of summaries, yielding either those that miss critical information or are mixed with irrelevant information. In the proposed model, the ACRL approach extends the Q-learning-based approach and achieves continuity in feedback through the iterative refinement of summaries by resorting to the dual structure with the actor generating summaries and the critic evaluating them. This dynamic adjustment leads to higher-quality summaries compared to the Q-learning approach in *Sharaff, Jain & Modugula (2022)*, mainly regarding ROUGE and BLEU scores, which have largely improved in the proposed model.

However, although *Mishra et al. (2022)* provides some adaptability to new domains, it relies heavily on large domain-specific training datasets & samples. Though this method performs very well when trained with data from a particular domain, it shows a drastic performance drop in zero-shot or few-shot learning scenarios. In contrast, the proposed meta-learning for zero-shot summarization will enable the model to generalize into new domains effectively using minimal samples of training data. During meta-training, it learns quick adaption to new tasks, which suits well the tasks for which large amounts of labeled data are not available. Also, in *Mishra et al. (2022)*, the inclusion of external

knowledge is missing due to which sometimes the ability is hampered while generating semantically rich summaries in specialized domains. The proposed model fills this gap by allowing domain-specific knowledge from external sources into the knowledge-enhanced transformer so that the summaries are contextually accurate and packed with deeper insights for semantic depth.

In a nutshell, though *Saini et al. (2023)*, *Sharaff, Jain & Modugula (2022)*, and *Mishra et al. (2022)* have contributed significantly to document summarization, each of them has specific limitations concerning hierarchical text processing, multimodal data incorporation, reinforcement learning-based optimization, new domain adaptation, and external knowledge incorporation. The proposed model overcomes these shortcomings on all fronts and presents a much stronger, more versatile, and effective system for high-quality summarization in various scenarios and datasets & samples. This is clearly reflected in the comparative improvement of ROUGE and BLEU scores on different benchmark datasets & samples.

Across all datasets, the proposed model consistently surpasses the performance of the established methods *Saini et al. (2023)*, *Sharaff, Jain & Modugula (2022)*, and *Mishra et al. (2022)*, underscoring its superior design and integration of advanced neural architectures. The significant advancements in ROUGE-2 and BLEU scores across the board highlight the model's enhanced ability to capture and reproduce not just the factual content but also the subtleties of language, narrative flow, and domain-specific nuances.

- **On textual summarization**: The CNN/Daily Mail and XSum results affirm the model's robustness in handling extensive narratives and distilling them into coherent, concise summaries.
- **On multimodal summarization**: The Multi30k dataset results showcase the model's adeptness at integrating and summarizing information from disparate sources, paving the way for richer, more informative content synthesis.
- **On knowledge integration**: The RDF2Text dataset performance illustrates how the incorporation of external knowledge can significantly enhance summary quality, offering deeper insights and contextual relevance.
- **On domain-specific adaptability**: Results from the custom domain-specific datasets demonstrate the model's exceptional zero-shot and few-shot learning capabilities, enabling it to quickly adapt to and accurately summarize content from previously unseen domains.

These results collectively underscore the effectiveness of the proposed integrated summarization model in addressing the challenges of automated summarization across a spectrum of scenarios and domains. Through its innovative combination of recursive neural networks, multimodal processing, reinforcement learning, meta-learning, and knowledge enhancement, the model sets a new standard for the generation of automated summaries, characterized by their depth, coherence, and contextual richness in real-time use case scenarios. Next, we discuss a practical use case of the proposed model, which will assist readers to understand the entire summarization process.

**Table 6   Output of ReTran.**

| Feature | Value |
|---|---|
| Input Text | "Recent breakthroughs in renewable energy technology..." |
| Processed Text | "Advancements in solar and wind energy have led to reduced costs..." |
| Summary (ReTran) | "Technological advancements in renewable energy have significantly lowered energy costs, promising a sustainable future." |

**Table 7   Output of multimodal transformer equipped with cross-modal attention.**

| Feature | Value |
|---|---|
| Input Text | "Recent breakthroughs in renewable energy technology..." |
| Input Image | "Image of solar panels" |
| Processed Text | "Solar panel advancements lead to lower costs..." |
| Processed Image | "Latest solar panel installations" |
| Summary (Multimodal) | "Recent solar panel advancements depicted in the latest installations promise significant cost reductions." |

## Example use case

To demonstrate the capabilities of our integrated summarization model, we present a series of examples showcasing the model's performance across its various components: ReTran, multimodal transformer with cross-modal attention, actor-critic reinforcement learning, meta-learning for zero-shot summarization, and knowledge-enhanced transformer for summarization. Each example is derived from processing the same initial dataset, which includes sample text and multimodal data, to illustrate how each component contributes to the final summary output. The dataset comprises a sample news article on renewable energy advancements, accompanied by an image of solar panels and metadata including the publication date and geographical information. The text highlights recent technological breakthroughs, the impact on energy costs, and future projections for renewable energy adoption. The image provides a visual representation of the latest solar panel installations, while the metadata indicates the article was published in California on March 15, 2023.

The ReTran component effectively distills the key points from the detailed text, focusing on the central theme of technological advancements and their impact on reducing energy costs is shows in Table 6

Leveraging both the textual and visual data, the multimodal transformer provides a nuanced summary that not only acknowledges the technological breakthroughs but also visually contextualizes the advancements through the latest solar panel installations in different scenarios is shows in Table 7.

The actor-critic component refines the initial summary by incorporating feedback to emphasize the most impactful aspects of renewable energy advancements: cost reduction and sustainability is shows in Table 8.

**Table 8  Output of actor-critic reinforcement learning.**

| Feature | Value |
|---|---|
| Initial Summary | "Technological advancements in renewable energy have significantly..." |
| Feedback Adjustments | "Emphasize cost reduction and sustainability" |
| Revised Summary (ACRL) | "Renewable energy advancements have dramatically cut costs and bolstered sustainability efforts." |

**Table 9  Output of meta-learning for zero-shot summarization.**

| Feature | Value |
|---|---|
| Unseen Domain Text | "Breakthroughs in biofuel production have been reported..." |
| Zero-Shot Summary (Meta) | "Innovations in biofuel technology hold promise for eco-friendly fuel alternatives and cost efficiency." |

**Table 10  Output of knowledge-enhanced transformer for summarization.**

| Feature | Value |
|---|---|
| Input Text | "Recent breakthroughs in renewable energy technology..." |
| External Knowledge Integration | "Renewable energy trends in California show a 30% adoption rate increase..." |
| Summary (Knowledge-Enhanced) | "In California, recent renewable energy advancements have surged adoption rates by 30%, indicating a shift towards sustainable power solutions." |

Despite being presented with text from an unseen domain (biofuels), the meta-learning component adeptly generates a coherent summary, highlighting the model's ability to adapt and summarize content from novel domains effectively is shows in Table 9.

By integrating external knowledge, the knowledge-enhanced transformer enriches the summary with specific data on renewable energy adoption rates in California, providing a detailed and contextually rich overview that underscores the significance of the advancements. The presented examples from Table 7 through Table 10 illustrate the multifaceted capabilities of our integrated summarization model. Each component— ReTran, the multimodal transformer, actor-critic reinforcement learning, meta-learning for zero-shot summarization, and the knowledge-enhanced transformer—plays a pivotal role in analyzing, interpreting, and synthesizing information from diverse data sources. Together, they contribute to generating summaries that are not only accurate and concise but also richly informative and contextually nuanced. The model's adaptability, as evidenced by its performance across various datasets and scenarios, positions it as a powerful tool for automated summarization tasks, capable of addressing the increasingly complex demands of information synthesis in the digital ages.

This integrated approach, combining the strengths of various advanced neural architectures and methodologies, showcases a significant leap forward in the field of automated summarization. Notably, the model's ability to integrate and leverage external

knowledge sources directly addresses the critical need for summaries that go beyond surface-level information, providing depth, context, and actionable insights for different scenarios. Moreover, its proficiency in handling multimodal data sources emphasizes the model's versatility and aligns with the evolving nature of digital content, which increasingly encompasses text, images, and other data types.

The adaptability demonstrated through zero-shot and few-shot learning capabilities signifies a remarkable advancement in the model's ability to generalize across domains. This is particularly crucial in an era where the volume and variety of information continue to expand, often outpacing the ability of traditional models to adapt to new or niche domains without extensive retraining. The use of actor-critic reinforcement learning for iterative refinement and optimization of summaries further enhances the model's performance, ensuring that the generated summaries are not only relevant and coherent but also aligned with user preferences and feedback. This iterative feedback loop, embodying the principles of reinforcement learning, ensures continuous improvement and adaptation of the summarization process, ultimately leading to higher quality outputs. In conclusion, the integrated summarization model presented in this study embodies a holistic and advanced approach to automated summarization. Its comprehensive design, combining recursive neural networks, multimodal processing, reinforcement learning, meta-learning, and knowledge enhancement, sets a new benchmark for the generation of automated summaries. Future work will focus on expanding the model's capabilities, exploring new data sources, further enhancing its adaptability to diverse domains, and optimizing its computational efficiency. As the landscape of digital information continues to evolve, so too will the demand for sophisticated summarization technologies, making the advancements demonstrated by this model not only relevant but essential for the future of information processing and dissemination.

## CONCLUSION & FUTURE SCOPE

The study introduces a comprehensive and innovative approach to multiple document summarization, integrating recursive transformer networks (ReTran), multimodal transformer with cross-modal attention, actor-critic reinforcement learning, meta-learning for zero-shot summarization, and knowledge-enhanced transformers. The model's performance was rigorously evaluated across various datasets, demonstrating its superior ability to generate concise, coherent, and contextually enriched summaries. Experimental results highlight the efficacy of the proposed model, with remarkable ROUGE-1, ROUGE-2, and ROUGE-L scores of 42.9, 20.5, and 39.4, respectively, on the CNN/Daily Mail dataset, along with a BLEU score of 42.3, showcasing significant advancements over existing methods. Similarly, on the Multi30k dataset, the model exhibits proficiency in processing and summarizing multimodal data, evident in scores of 61.2 (ROUGE-1), 42.8 (ROUGE-2), 59.5 (ROUGE-L), and 62.3 (BLEU), indicating improved integration of visual and textual information into cohesive summaries. Additionally, the model demonstrates exceptional adaptability and efficiency in learning from limited examples, as evidenced by its performance on custom domain-specific datasets, achieving high ROUGE and

BLEU scores. Looking ahead, future exploration could focus on additional knowledge sources, cross-lingual summarization, domain-specific optimization, interpretability and explainability, and scalability and efficiency. In conclusion, the proposed model represents a significant advancement in automated summarization, offering accurate, concise, and contextually rich summaries, with the potential to address evolving challenges in information synthesis.

### Funding
The authors received no funding for this work.

### Competing Interests
The authors declare there are no competing interests.

### Author Contributions
- Sunilkumar Ketineni conceived and designed the experiments, performed the experiments, analyzed the data, performed the computation work, prepared figures and/or tables, authored or reviewed drafts of the article, and approved the final draft.
- Sheela Jayachandran performed the experiments, performed the computation work, prepared figures and/or tables, authored or reviewed drafts of the article, and approved the final draft.

### Data Availability
The data for each DUC workshop is available at: https://duc.nist.gov/data.html.

### Supplemental Information
Supplemental information for this article can be found online at http://dx.doi.org/10.7717/peerj-cs.2463#supplemental-information.

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
