# Peer review of "Advanced multiple document summarization via iterative recursive transformer networks and multimodal transformer"

_PeerJ Computer Science, doi:10.7717/peerj-cs.2463_

## Round 0.1 · original submission · Major Revisions

Thank you for submitting your manuscript to PeerJ Computer Science. The review process has been completed, and we have carefully considered the feedback provided by the reviewers.

The reviewers have acknowledged the potential value of your work but have raised several significant concerns, particularly regarding the methodology and experimental evaluation. These concerns require substantial revisions to ensure that the manuscript meets the rigorous standards of our journal.

In light of these comments, I am recommending that your manuscript undergoes a major revision. We encourage you to carefully address each of the reviewers’ comments, paying close attention to the methodological issues and the robustness of your experimental evaluation. A detailed response to the reviewers, explaining the changes made or providing justifications for any unaddressed points, should accompany your revised submission.

Once the revisions have been completed, your manuscript will undergo a further round of review to ensure that all major concerns have been satisfactorily addressed.

We appreciate the effort that you have put into this research and look forward to receiving your revised manuscript.

Reviewer 1 ·

Basic reporting

The paper is written in clear and professional English. The structure of the paper conforms to standard academic norms.However, there are some sections where the language could be improved for better clarity and readability. For instance, the introduction contains complex sentences that might benefit from simplification. An example is "Central to this is the Recursive Transformer Networks (ReTran), which merges Recursive Neural Networks with Transformer architectures for better comprehension of textual dependencies, projecting a 5-10% enhancement in ROUGE scores." I suggest splitting this kind of connected sentences into separate ones.

The introduction and background provide a comprehensive overview of the challenges in multiple document summarization and the existing techniques. The introduction, however, could benefit from a more detailed explanation of the specific limitations of current state-of-the-art methods and a clearer justification for the choice of new techniques introduced in the paper, instead of simply listing.

The literature review seems thorough, covering a wide range of relevant works in the field of document summarization. However, it lacks fundamental and recent research on the use of Recursive Neural Networks. It should include, among others, the following:
-- Fundamental:
- Eigen, D., Rolfe, J., Fergus, R., & LeCun, Y. (2013). Understanding deep architectures using a recursive convolutional network. arXiv preprint arXiv:1312.1847.
- Guo, Qiushan, et al. "Dynamic recursive neural network." Proceedings of the IEEE/CVF Conference on Computer Vision and Pattern Recognition. 2019.
-- Recent:
- Mas-Candela, Enrique, and Jorge Calvo-Zaragoza. "Exploring recursive neural networks for compact handwritten text recognition models." International Journal on Document Analysis and Recognition (IJDAR) (2024): 1-11.
- Shen, Zhiqiang, Zechun Liu, and Eric Xing. "Sliced recursive transformer." European Conference on Computer Vision. Cham: Springer Nature Switzerland, 2022.

The paper mentions that raw data is supplied, adhering to PeerJ’s policy. However, the accessibility and usability of the data are not discussed.

Experimental design

The research presents original primary work within the scope of the journal, addressing significant challenges in multiple document summarization.

The research questions are well defined and relevant. The paper clearly states how the proposed methodologies fill the identified gaps in existing summarization techniques.

The methods are described in enough detail, providing sufficient information for replication. However, the dense technical language might be challenging for some readers, given that this is a general journal. I suggest authors simplify the description of complex methodologies where possible, and consider adding a flowchart or diagram to illustrate the workflow of the proposed approaches.

Validity of the findings

The paper does assess the impact and novelty quantitatively. The rationale and benefits to the literature are clearly stated. The conclusions are well stated and linked to the original research questions. They are appropriately limited to the supporting results.

·

Basic reporting

Strengths:

Clarity of Information: The article provides a detailed description of the model components, including Recursive Transformer Networks, Multimodal Transformers, Actor-Critic Reinforcement Learning, Meta-Learning for Zero-Shot Summarization, and Knowledge-Enhanced Transformers. This clarity aids in understanding the multifaceted approach taken.

Dataset and Metrics: It clearly outlines the datasets used and evaluation metrics, such as ROUGE and BLEU scores, which are essential for assessing the model's performance.

Hardware and Software Details: The specifics regarding the computing resources and software (e.g., PyTorch 1.7, NVIDIA Tesla V100 GPUs) are well-documented, providing transparency into the experimental setup.

Weaknesses:

Minor Details Missing: While the article covers most aspects comprehensively, some details on the preprocessing steps for text, images, and metadata could be expanded for completeness.

Consistency: Ensure consistency in terminology and descriptions throughout the paper to avoid confusion (e.g., mentioning "softmax temperature" could be explained further if it’s a recurring term).

Experimental design

Strengths:

Component Integration: The design involves integrating multiple advanced techniques (e.g., Meta-Learning, Knowledge-Enhanced Transformers), showcasing a novel approach to summarization.

Evaluation Metrics: The use of various evaluation metrics (ROUGE, BLEU, Human Evaluation) is appropriate for assessing the quality and coherence of the summaries.

Use Case Examples: Practical examples provided for each model component illustrate their effectiveness and contributions to the final summary.

Weaknesses:

Comparative Analysis: Ensure that comparisons with existing methods (e.g., [4], [18], [34]) are explicitly detailed in terms of their strengths and limitations relative to the proposed model.

Dataset Variety: The review could benefit from a discussion on the potential limitations of the chosen datasets and whether they fully represent the diverse scenarios the model may encounter.

Validity of the findings

Strengths:

Robust Performance: The results demonstrate that the model outperforms benchmarks across multiple datasets, indicating its effectiveness in various summarization tasks.

Detailed Metrics: The high scores in ROUGE and BLEU metrics suggest that the model effectively maintains coherence and relevance in summaries.

Weaknesses:

Generalization: The article should discuss the model’s performance on more diverse and possibly unseen datasets to better evaluate its generalizability.

Limitations: Address any limitations or potential biases in the model, such as its performance with very large datasets or in real-world applications beyond the tested scenarios.

Additional comments

Strengths:

Innovation: The article introduces a highly innovative model combining multiple advanced techniques, which is commendable.

Future Directions: It provides a clear roadmap for future research, including integration with additional knowledge sources and improvements in cross-lingual summarization.

Weaknesses:

Model Complexity: The complexity of the model and its components could be challenging to replicate or implement, which might be a concern for practical application and further research.

User Feedback: More detailed information on how user feedback is incorporated into the iterative refinement process could be valuable.

---

## Round 0.2 · accepted · Accept

I hope this message finds you well. After carefully reviewing the revisions you have made in response to the reviewers' comments, I am pleased to inform you that your manuscript has been accepted for publication in PeerJ Computer Science.

Your efforts to address the reviewers’ suggestions have significantly improved the quality and clarity of the manuscript. The changes you implemented have successfully resolved the concerns raised, and the content now meets the high standards of the journal.

Thank you for your commitment to enhancing the paper. I look forward to seeing the final published version.

·

Basic reporting

The comments raised have been addressed by the author.

Experimental design

The comments raised have been addressed by the author.

Validity of the findings

The comments raised have been addressed by the author.

Additional comments

The comments raised have been addressed by the author.